# Critical Role of the Cortical Alveolus Protease Alveolin in Chorion Hardening In Vivo at Medaka Fertilization

**DOI:** 10.3390/biom13010146

**Published:** 2023-01-10

**Authors:** Bo Fu, Di Wu, Shigeki Yasumasu, Masaya Hane, Chihiro Sato, Ken Kitajima

**Affiliations:** 1Bioscience and Biotechnology Center, Nagoya University, Chikusa, Nagoya 464-8601, Japan; 2Graduate School of Bioagricultural Sciences, Nagoya University, Chikusa, Nagoya 464-8601, Japan; 3Institute of Glyco-core Research, Nagoya University, Chikusa, Nagoya 464-8601, Japan; 4Department of Materials and Life Sciences, Faculty of Science and Technology, Sophia University, Tokyo 102-8554, Japan

**Keywords:** alveolin, cortical alveolus proteinase, chorion hardening, fertilization, medaka, transglutaminase, zona pellucida

## Abstract

Alveolin is a cortical alveolus proteinase that is secreted in the perivitelline space (PVS) at fertilization to act on the chorion. Purified alveolin is known to induce chorion hardening in vitro by processing zona pellucida B (ZPB), a major chorion component. However, in vivo function of alveolin remains unclear; thus, in this study, the effects of *alveolin* efficiency (*Alv*^−/−^) at the organism level were investigated using the medaka, *Oryzias latipes*. The *Alv*^−/−^ fertilized eggs were mechanically fragile; however, they developed normally and left offspring as long as they were carefully handled before hatching. A mechanical press test showed that the *Alv*^−/−^ fertilized eggs were six times more fragile than the wild-type eggs. They were 35% larger owing to the enlarged PVS, 34% thinner, and permeable to even 10 kDa FITC-dextran. These results are consistent with the transmission electron microscopy observation that the periphery of the inner layers was highly porous in the *Alv*^−/−^ chorion. In chorion hardening, the alveolin-mediated processing of ZPB and the transglutaminase (TGase)-mediated crosslinking of chorion components are the key steps. This study was the first to show that alveolin also processed TGase concomitantly with ZPB, which greatly facilitated the crosslinking. Thus, alveolin was concluded to be the primary trigger for chorion hardening in vivo. Furthermore, fertilization in a balanced salt solution could partially improve the impaired chorion hardening of the *Alv*^−/−^ eggs fertilized in water, probably through an alveolin-independent mechanism.

## 1. Introduction

In most animals, the embryo is covered by the egg envelope, an extracellular proteinaceous layer that is differently named depending on the organism: zona pellucida (ZP) in mammals, perivitelline layer in birds, and vitelline envelope in other animals, though often called chorion in fish [1,2]. The fish chorion is mechanically fragile before fertilization; however, upon fertilization, it changes to a stiff envelope, i.e., the fertilization envelope (FE), through the chorion hardening process. This process establishes a slow and complete polyspermy block by occluding the micropyle, a small pore in the chorion through which the sperm passes [3]. It also enables not only mechanical and microbial protection of embryos [4,5], but also constant maintenance of their surrounding environment [6,7] by isolating them from the external environment. The formation of FE also allows embryos to regulate their respiration, excretion of metabolic wastes, and adjustment of ionic and osmotic balances in the FE [8]. This is the formation of fertilization membranes in sea urchins and amphibians [9,10] and the zona reaction in mammals [11].

The egg envelope mainly consists of ZP glycoproteins, which are characterized by the presence of ZP domains that are highly conserved in multicellular eukaryotes from Cnidarians to humans [12]. In medaka fish, *Oryzias latipes*, zona pellucida Bs (ZPBs and choriogenin H and H minor) and zona pellucida C (ZPC and choriogenin L) are present as the major ZP glycoproteins [13]. The medaka egg envelope or chorion is composed of two layers: a thick inner layer and a thin outer layer; the former contains ZPBs and ZPC as major proteins [14]. During chorion hardening at fertilization, ZPB is proteolytically hydrolyzed by alveolin (Alv), an oocyte-specific astacin family protease [15,16], and cross-linked to ZPC through the ε-(γ-glutamyl) lysine linkage formed by a transglutaminase (TGase) to form ZPB–ZPC polymers that are insoluble high-molecular-weight complexes [17,18]. The processing and polymerization of chorion proteins are considered to cause chorion hardening because these changes are reproduced in vitro using isolated chorion and Alv [17,18]. However, the mechanism by which Alv and TGase are involved in chorion hardening remains unclear. Thus, to unveil the molecular mechanism and biological significance of chorion hardening at the organism level, *alveolin* knockout (*Alv*^−/−^) medaka was generated using clustered regularly interspaced short palindromic repeats (CRISPR)/CRISPR-associated protein 9 (Cas9) gene editing techniques. The data of this study have demonstrated that Alv is critically involved in chorion hardening and that chorion hardening is important for mechanical strength and the permeability of chorions during fertilization, as well as embryonic development before hatching.

## 2. Materials and Methods

### 2.1. Medaka

The orange–red strain of medaka fish, *Oryzias latipes*, was kindly gifted by Dr. Masahiko Hibi and Dr. Hisashi Hashimoto (Graduate School of Science, Nagoya University). The fish stocks were maintained in 16 L tanks with a water circulating system at 26 °C in a 14/10 h day/night cycle. The development and phenotype of the medaka fish were observed under a microscope (Olympus SZX12 DP80, Tokyo, Japan). The balanced salt solutions for medaka (BSS) were prepared as previously described: the salt solution contains 0.1 M NaCl, 5 mM KCl, 1 mM CaCl_2_, and 0.6 mM MgSO_4_ [19,20].

### 2.2. Single-Guide RNAs and Cas9 Synthetic RNA

A single copy of the *alveolin* gene exists in medaka (LOC100049214) [15]. Based on this sequence, guide RNAs were designed using the software tool “Search for CRISPR target site with micro-homology sequences” (http://viewer.shigen.info/cgi-bin/crispr/crispr.cgi (accessed on 1 April 2020)) for the prediction of unique target sites and a pattern match tool software (http://viewer.shigen.info/medakavw/crisprtool/ (accessed on 1 April 2020)) to search for possible off-target sites in the genome [21]. The target sequence (CTCCAGTTCCCTCCACACAGG; the underlined region is a proto-spacer adjacent motif sequence) was selected from exon 2. A pair of oligonucleotides (Hokkaido System Science Co., Ltd., Hokkaido, Japan), 5′-TAGGCTCCAGTTCCCTCCACAC-3′ (sense) and 5′-AAACGTGTGGAGGGAACTGGAG-3′ (antisense), were annealed at 10 μM and the annealed DNA was cloned into the BsaI-digested pDR274 plasmid (#42250, Addgene) according to the manufacturer’s instructions. The constructed plasmid was then linearized by digestion with DraI and used as a template for in vitro transcription with the AmpliScribe T7-Flash Transcription Kit (Epicentre Technologies Corporation of Madison, WI, USA). For Cas9, capped mRNA was synthesized using NotI-digested pCS2 + hSpCas9 plasmid (#51815, Addgene, Watertown, MA, USA) as a template and the mMessage mMachine SP6 Transcription Kit (Thermo Fisher Scirntific, Tokyo, Japan). The primers used were 5′-GCAGGATCCGCCACCATGGACTATAAGGAC-3′ (sense) and 5′-AGTTCTAGATTACTTTTTCTTTTTTGCCTGGC-3′ (antisense). Both the synthesized capped Cas9 RNA and single-gRNA were purified using an RNeasy Mini Kit (Qiagen, Hilden, Germany).

### 2.3. Generation of Alveolin-Deficient (Alv^−/−^) Medaka Strain Using the CRISPR-Cas9 System

Approximately 2–4 nL of a mixture of gRNA (25 ng/μL) and Cas9 mRNA (100 ng/μL) was injected into fertilized eggs before the first cleavage by an MMN-8 micromanipulator (Narishige, Tokyo, Japan). The injected embryos (F0) were maintained in water at 26 °C in an Incubate Box M-260F (Taitec, Saitama, Japan). The RNA-guided engineered nucleases-injected fish were mated with wild-type (WT) fish of the orange–red strain to obtain heterozygous (F1) fish with a pair of normal and mutated alleles with a mutation at the target locus. The mutations in each embryo were analyzed using the heteroduplex mobility assay (HMA) (see Section 2.4). The offspring were then crossbred to obtain an individual founder fish.

### 2.4. Genotyping

Genomic DNA was individually prepared from embryos at more than 3 days post-fertilization (dpf). Each embryo was lysed in 25 μL of alkaline lysis buffer containing 25 mM NaOH and 0.2 mM ethylenediaminetetraacetic acid (EDTA) (pH 8.0) and incubated at 95 °C for 15 min after being broken with forceps. The lysate was neutralized with 25 μL of 40 mM Tris-HCl (pH 8.0) and used as the genomic DNA. Heterozygous/homozygous genotypes were identified using the HMA [22,23,24] and/or nucleotide sequence analysis. A 296-bp fragment containing the entire genomic target sequence of the *alveolin* gene was amplified using the primers ALV-exon-3-F (CCTGGGTCTATGCTGTGCTATG) and ALV-exon-3-R (CTGCAAGAACCCACCTTTACC). The reaction mixture contained 1 µL of genomic DNA as a template, 1× polymerase chain reaction (PCR) buffer for KOD-Plus-Neo, 0.2 mM of each dNTP, 1.5 mM of MgSO_4_, 0.2 µM of each primer, and 0.02 unit of KOD-Plus-Neo (TOYOBO) in a total volume of 10 µL. The cycling conditions were as follows: one cycle at 94 °C for 2 min, followed by 35 cycles at 98 °C for 10 s, 67.5 °C for 30 s, and 68 °C for 8 s. The resulting amplicons were electrophoresed on either 15% polyacrylamide gels or 2% agarose gels. Alternatively, the PCR amplicons were sequenced using ALV-exon-3-F as a primer.

### 2.5. Reverse Transcriptase–Polymerase Chain Reaction

The total RNA was extracted from various organs excised from adult medaka fish using TRI REAGENT^®^ LS (Thermo Fisher Scirntific, Tokyo, Japan). The cDNA of each sample was reverse transcribed using ProtoScript II (New England Biolabs, Inc, Ipswich, MA, USA), with a random hexamer primer. RThe everse transcriptase–polymerase chain reaction was performed using the following primers: for *alveolin* gene, 5′-TGCAGGATGTGGGGGCTCCT-3′ (forward) and 5′-TCAGTATTTGGAGCCACAGC-3′ (reverse) and, for *β-actin* gene, 5′-CCCAGAAAGACAGCTACGTA-3′ (forward) and 5′-TGATCTTCATGGTGGATGGG-3′ (reverse).

### 2.6. Estimation of Yolk and Perivitelline Space Volumes

The *Alv*^+/−^ medaka were mated to obtain *Alv*^+/+^, *Alv*^+/−^, and *Alv*^−/−^ embryos at 4 hpf. The fertilization rate was 90–100% and the hatching rate was 90–100% for *Alv*^+/+^ and *Alv*^+/−^ fry and around 20% for *Alv*^−/−^ fry on the culture in water. The genotyping was performed after morphological observations. The morphologies of the 4 hpf embryos and 9 dpf fry just after hatching were observed using an Olympus SZX12 microscope and the photographs were taken using an Olympus DP80 camera. The images were processed using ImageJ software (Version 2, https://imagej.nih.gov/index.html (accessed on 1 April 2020)). The egg and yolk diameters were measured and the whole egg volume, yolk volume, and perivitelline space (PVS) volume were calculated using the volume formula, V = 4/3 πr^3^. The volume of the PVS was equal to the yolk volume subtracted from the egg volume.

### 2.7. Determination of the Chorion Toughness

The toughness of the chorion at given time intervals after fertilization was measured as resistance against the egg-crushing force using an ordinary dual pan balance. The egg at 0–140 min post-fertilization (mpf) was placed at the center of a balance pan under the blunt tip of a glass bar fixed on an iron stand. Water was added up to 10 mL and then Japanese coins were added one by one to the other pan until the egg was crushed. The total weight of the water and coins required to crush the egg was recorded as an index of chorion toughness.

### 2.8. Permeability Test of Eggs

The eggs at 3 hpf and 1 dpf with *Alv*^+/+^ and *Alv*^−/−^ genotypes were incubated in 5 mg/mL of FITC-dextran (10 kDa) (Cosmo Bio Co., Ltd., Tokyo, Japan) for 1 h and then washed three times with water or BSS. The fluorescence of each group was observed using an Olympus SZX12 microscope and Olympus DP80 camera system.

### 2.9. Histological Analysis of Chorion

The 1 dpf fertilized eggs of *Alv*^+/+^ and *Alv*^−/−^ medaka were treated with Davidson’s fixative solution containing formalin (37%), acetic acid, glycerin, ethanol, and water (2:1:1:3:3 *v*/*v*/*v*/*v*/*v*, respectively). The eggs were then dehydrated in graded ethanol (80–100%), cleared with Clear-rite 3, and embedded in paraffin. Paraffin sections of 5 μm thickness were cut using a manual rotary microtome. The histological structure of the chorion was visualized using hematoxylin and eosin staining. The photographs were obtained using an Olympus SZX12 microscope and Olympus DP80 camera system. The thickness of the chorions was also observed directly, the chorions were manually isolated from embryos with sharp tweezers, washed with water, and cut using a scalpel. The thicknesses of the chorions were measured using an Olympus SZX12 microscope and Olympus DP80 camera system.

### 2.10. Transmission Electron Microscopy

The fertilized chorions were fixed in 2% paraformaldehyde and 2% glutaraldehyde in 0.1 M sodium cacodylate buffer (pH 7.4) overnight at 4 °C and then post-fixed with 2% osmium tetroxide in 0.1 M cacodylate buffer at 4 °C for 3 h. After dehydration using a graded ethanol series, the samples were placed in propylene oxide and embedded in 100% resin (Quetol-812; Nisshin EM Co., Tokyo, Japan). Ultrathin sections (70 nm) cut with a diamond knife using an ultramicrotome (Ultracut UCT; Leica, Vienna, Austria) were stained using uranyl acetate and lead stain solution (Sigma-Aldrich Co., Tokyo, Japan) and examined using transmission electron microscopy (JEM-1400Plus; JEOL Ltd., Tokyo, Japan). The digital images (3296 × 2472 pixels) were captured using a CCD camera (EM-14830RUBY2; JEOL Ltd., Tokyo, Japan). All the procedures after the fixation were consigned to the Tokai Electron Microscopic Analysis, Co., Ltd., Nagoya, Japan.

### 2.11. Preparation of Anti-Hardening Transglutaminase Antibody

The cDNA for the medaka orthologue of hardening transglutaminase (TGase) that was originally purified from rainbow trout, *Oncorhynchus mykiss* [25] was cloned (Yasumasu S. to be published elsewhere). The cDNA fragment for the C-terminal domain of the hardening TGase was amplified using PCR and cloned into the pET3c expression vector. The recombinant protein was expressed in *E. coli* (BL21) and the obtained inclusion body was dissolved in 50 mM Tris–HCl (pH 8.0), 8 M urea, and 1 mM EDTA and subjected to a Ni-NTA Agarose column (Thermo Fisher Scientific, Tokyo, Japan). The purified proteins were used as an immunogen to raise antibodies in mice. After immuno-injection was performed once a week four times, the serum was collected for isolation of IgG antibody.

### 2.12. Biochemical Analysis of Chorion by Coomassie Brilliant Blue R-250 Staining and Western Blotting

The fertilized chorion was isolated and washed with 10 mM EDTA in the phosphate-buffered saline (PBS). Unfertilized chorions were isolated from spawning female ovaries, crushed with sharp tweezers in 10 mM EDTA in PBS, and washed with the same solution. The isolated chorion was denatured in 20 µL of 1% SDS at 100 °C for 3 min. The chorion extract was mixed with 4 µL of 6 X Laemmli buffer containing 5% 2-mercaptoethanol and was subsequently heated at 100 °C for 5 min. Typically, the protein concentration in the 24 µL-extract from two chorions was 0.5–1 mg/mL. For Coomassie Brilliant Blue R-250 (CBB) staining, the prepared samples were separated by 7% SDS-PAGE gels and stained with CBB. For Western blotting, the prepared samples were separated on 10% SDS-PAGE and blotted onto a polyvinylidene fluoride membrane. The membrane was blocked with PBS with 0.1% Tween 20 (PBST) containing 1% skim milk at 25 °C for 1 h and then incubated with the primary antibody, anti-hardening TGase antibodies (1/1000 dilution), at 4 °C overnight. After washing with PBST, the membrane was incubated with a secondary antibody, horseradish peroxidase (HRP)-conjugated anti-mouse IgG + M antibody (1/5000 dilution), at 37 °C for 1 h. After washing with PBST, the immunoblots were developed using an enhanced chemiluminescence Western blotting detection reagent and detected using light capture.

### 2.13. Statistical Analysis

The data were processed to show the mean and the standard deviation for normality of the data set. The statistical analysis was performed using Student’s *t*-test. The statistical significance was set at *p* < 0.05. All the data were processed using the Excel software Version 16 (accessed on 1 April 2020).

## 3. Results

### 3.1. Alv^−/−^ Medaka Normally Develop

To investigate the in vivo function of Alv, the *Alv*^−/−^ medaka strains were established using the CRISPR-Cas9 system. Because of a 4-bp insertion at a position within the pro-domain of Alv at exon 2 (Figure 1A), the mutated allele encoded a 69 amino-acid protein without the catalytic domain (Figure 1B), suggesting that no functional Alv protein was expressed in the *Alv*^−/−^ medaka. The genotypes of *Alv*^−/−^, *Alv*^+/*−*^, and *Alv*^+/+^ medaka could be classified by two rounds of HMA assay of the amplicon including the 4 bp insertion site; i.e., at the first HMA, *Alv*^+/*−*^ provided slowly migrated bands, while *Alv*^−/−^ and *Alv*^+/+^ rapidly migrated together. *Alv*^−/−^ and *Alv*^+/+^ could be differentiated from each other at the second HMA by mixing with the *Alv*^+/+^-derived amplicon (Figure 1C).

As described below, the most prominent feature of *Alv*^−/−^ medaka was the mechanical fragility of fertilized eggs. Therefore, the eggs after fertilization had to be carefully handled. However, as a whole, the *Alv*^−/−^ medaka could normally develop, hatch, grow up, and reproduce to produce the next generation, except for small but significant morphological differences in fry immediately after hatching. More specifically, in just-hatched fry, the body length and the yolk volume of *Alv*^−/−^ fry were 18% shorter (3.6 mm for *Alv*^−/−^ vs. 4.4 mm for *Alv*^+/−^) and 50% larger (0.24 mm^3^ for *Alv*^−/−^ vs. 0.086 mm^3^ for *Alv*^+/−^) than those of *Alv*^+/−^ fry, respectively (Figure 1D). A larger yolk volume suggests a slower incorporation rate of yolk nutrients into fry bodies, which might induce growth retardation and shorter body length. However, these features of *Alv*^−/−^ fry appear to be overcome when the fry start intaking food.

### 3.2. Alv^−/−^ Medaka Produce Soft Chorion Eggs

Fertilized eggs are usually attached to the genital pore (Figure 2A) immediately after spawning. When the fertilized eggs from the *Alv*^−/−^ parent pair were collected at 1–2 h post-fertilization (hpf), 90% of the spawned eggs attached to the medaka were found to be destroyed similar to ghost envelopes that were devoid of yolk and embryos (Figure 2B). Therefore, the fertilized eggs from *Alv*^−/−^ were carefully collected using a pipette by careful sucking, as suggested in the NBRP protocol (https://medaka-book.org/contents/chapter03/m03-02c.html (accessed on 1 November 2022)). As shown in Figure 2C, the chorions of the *Alv*^−/−^ eggs were soft and deformed when gently pricked using a forceps tip. In contrast, those of the *Alv*^+/+^ and *Alv*^+/−^ eggs were tough and robust against the similar mechanical pressure. The chorion phenotype of the *Alv*^−/−^ eggs was thus defined as “soft chorion”, compared with the “tough chorion” of the *Alv*^+/+^ and *Alv*^+/−^ eggs. To test how female and male parents of the three genotypes affected the soft chorion phenotype, reciprocal crossing experiments were performed (Figure 2D). The soft chorion was observed only when the *Alv*^−/−^ female, but not male, was used for the mating, indicating that the soft chorion phenotype depends on *Alv*^−/−^ genotype in a female-specific manner. When the expression of the *alveolin* gene was examined in various male and female organs (Figure 2E), it was found to be exclusively ovary specific, as previously described [16]. The ovary-specific expression of *alveolin* is consistent with the results of the reciprocal crossing experiment.

### 3.3. Characterization of Soft Chorion Eggs from Alv^−/−^ Medaka

#### 3.3.1. Size of Fertilized Eggs

When the fertilized eggs were observed from a top view, which was obtained by the stereoscope in a normal position, the *Alv*^−/−^ eggs had larger chorions than the *Alv*^+/+^ and *Alv*^+/−^ eggs (Figure 3A, top view). This observation became obvious when the eggs were observed from a side view, which was attainable using a typical optical microscope that was laid down by 90° (Figure 3A, side view). When the PVS and yolk volumes were measured, the PVS volume of *Alv*^−/−^ eggs was 1.7 times larger than those for *Alv*^+/+^ and *Alv*^+/−^ (0.64 mm^3^ for *Alv*^−/−^ vs. 0.39 and 0.37 mm^3^ for *Alv*^+/+^ and *Alv*^+/−^, respectively) (Figure 3A, right) and the yolk volume of *Alv*^−/−^ eggs was not significantly different than those of *Alv*^+/+^ and *Alv*^+/−^ eggs (0.63 mm^3^ for *Alv*^−/−^; 0.55 and 0.54 mm^3^ for *Alv*^+/+^ and *Alv*^+/*−*^, respectively; *p* > 0.05 in both cases) (Figure 3A, right). These data show that *Alv*^−^^/*−*^ fertilized eggs were larger than *Alv*^+/+^ and *Alv*^+/*−*^ eggs because of the enlarged PVS volumes. Since the chorion is known as a semi-permeable membrane involved in the osmotic pressure of the egg [2,26], the PVS and yolk volumes were also compared between eggs that were fertilized in water and BSS, an isotonic solution to the egg (Figure 3B). The yolk volume was the same between the fertilized eggs in water and BSS (0.38 vs. 0.41 mm^3^). On the other hand, the PVS volume of *Alv*^−/−^ eggs in water was 1.5 time larger than that in BSS (0.56 vs. 0.38 mm^3^). The results showed that the PVS volume of the *Alv*^−^^/*−*^ eggs was reciprocal to the salinity, consistent with the fact that the chorion of the *Alv*^−^^/*−*^ eggs retains the semi-permeable property after fertilization.

#### 3.3.2. Mechanical Strength

To evaluate the mechanical strength of the soft chorion eggs, the toughness of the chorion was measured using an Iwamatsu’s device with a dual pan balance. A schematic representation of the device is shown in Figure 4A. Chorion hardening of eggs in water is known to proceed in a time-dependent manner [27]. As expected, the toughness of the chorion of the WT eggs that were fertilized and incubated in water (designated *Alv*^+/+^ eggs in water) increased in proportion to time after fertilization and the value was 90 g at 120 min post-fertilization (mpf) (Figure 4B, *Alv*^+/+^ eggs in water). In contrast, the *Alv*^−/−^ eggs that were fertilized and incubated in water (*Alv*^−/−^ eggs in water) were easily crushed by the 1–2 g for the first 150 mpf (Figure 4B, *Alv*^−/−^ eggs in water). Interestingly, the toughness of the chorion of *Alv*^−/−^ eggs that were fertilized and incubated in BSS (*Alv*^−/−^ eggs in BSS) moderately increased as the post-fertilization time increased, with a value of 32 g at 120 mpf (Figure 4B, *Alv*^−/−^ eggs in BSS). The toughness of the chorion of three types of eggs, i.e., *Alv*^+/+^ eggs in water, *Alv*^−/−^ eggs in water, and *Alv*^−/−^ eggs in BSS, was compared at 3 hpf (Figure 4C). The values were 118, 19.7, and 70.1 g for the *Alv*^+/+^ eggs in water, *Alv*^−/−^ eggs in water, and *Alv*^−/−^ eggs in BSS, respectively. Thus, the toughness value for *Alv*^−/−^ eggs was one-sixth that of the WT eggs. Therefore, *Alv*^−/−^ eggs prominently show a soft chorion phenotype. Notably, the toughness value of *Alv*^−/−^ eggs in BSS was 3.5 time higher than that of *Alv*^−/−^ eggs in water. The soft chorion phenotype of *Alv*^−/−^ eggs was partially rescued when the eggs were incubated in BSS instead of water. Although 90% of the fertilized *Alv*^−/−^ eggs from the *Alv*^−/−^ medaka pair in water were crushed during collection (Figure 2B), most fertilized *Alv*^−/−^ eggs could be collected intact when fertilization was performed in BSS.

#### 3.3.3. Permeability

The effects of *alveolin* deficiency on the permeability of the chorion of fertilized eggs in water were examined. The 3 hpf fertilized eggs were collected from parent pairs of *Alv*^+/+^, *Alv*^+/−^, and *Alv*^−/−^ genotypes. The fertilized eggs were incubated in 5 mg/mL of 10 kDa FITC-dextran for 1 h and the fluorescence in the PVS was observed after three washes with water (Figure 5A). The fluorescence was observed only in the fertilized eggs from *Alv*^−/−^-female and not those from *Alv*^+/+^- or *Alv*^+/−^-female. These data indicate that the permeability of the *Alv*^−/−^ chorion is increased enough to allow 10 kDa molecules to penetrate, compared with that of the *Alv*^+/+^ and *Alv*^+/−^ chorions. The effects of fertilization and culture in BSS on chorion permeability were also examined using four types of eggs, i.e., *Alv*^+/+^ eggs in water, *Alv*^+/+^ eggs in BSS, *Alv*^−/−^ eggs in water, and *Alv*^−/−^ eggs in BSS, which were collected at two different lengths of time, i.e., 3 hpf and 1 day post-fertilization (dpf) (Figure 5B). The data showed that the *Alv*^−/−^, but not *Alv*^+/+^, chorion was permeable to 10 kDa FITC-dextran, irrespective of whether the eggs were fertilized in water or BSS. In contrast, chorion toughness is, at least in part, affected by the BSS. These results indicate that the increased chorion permeability was due to *alveolin* deficiency.

### 3.4. Morphological Observations of the Chorion of Alv^−/−^ Eggs

#### 3.4.1. Light Microscopy

The histological observations of chorions isolated from fertilized eggs of *Alv*^+/+^ and *Alv*^−/−^ genotypes were performed (Figure 6A,B). The average thickness values of the sectioned chorions from *Alv*^+/+^ and *Alv*^−/−^ eggs were significantly different, i.e., 14 and 9.2 μm (*p* < 0.05), respectively (Figure 6A). Thus, the *Alv*^−/−^ chorion was thinner than that of the *Alv*^+/+^ chorion. As previously reported [27], the chorion consists of a multiple-layer structure (Figure 6B, left). The layer numbers of *Alv*^+/+^ and *Alv*^−/−^ chorions were on average 7.2 and 7.6, respectively (Figure 6B, right). However, the difference was not statistically significant. These results suggest that *Alv*^−/−^ eggs had thin chorions without changing their basic layer structure.

#### 3.4.2. Transmission Electron Microscopy

The chorion structure was observed using transmission electron microscopy. The chorion of each genotype retained a similar multiple-layer structure and each layer consisted of alternating pairs of a thin electrodense and a thick electrolucent layer (Figure 6C), which are the same images as previously reported for the WT chorion [28]. This is a common feature between *Alv*^+/+^ and *Alv*^−/−^ chorions, which were also observed under the light microscopy (Figure 6A,B). Notably, the periphery of the inner layer, which is shown by red squares in the upper panels in Figure 6C, contained disordered, multiple porous structures in the *Alv*^−/−^ chorion (Figure 6C, *Alv*^−/−^ eggs in water, *lower*), compared with the same parts in the *Alv*^+/+^ chorion (Figure 6C, *Alv*^+/+^ eggs in water, *lower*). The area of the disordered, multiple porous structure occupied approximately 50% of the whole inner layer of the chorion (Figure 6C, *Alv*^−/−^ eggs in water, *lower*). A similar disordered structure was also observed in the *Alv*^−/−^ chorion of the eggs that were fertilized and cultured in BSS (Figure 6C, *Alv*^−/−^ eggs in BSS, *lower*); however, approximately 20–30% was occupied by the disordered area, which was less prominent than that of the chorion of *Alv*^−/−^ eggs fertilized in water (Figure 6C, *Alv*^−/−^ eggs in water, *lower*). Because increasing chorion fragility of the eggs was observed across *Alv*^+/+^ eggs in water, *Alv*^−/−^
*eggs* in BSS, and *Alv*^−/−^
*eggs* in water (Figure 6C), the higher proportion of the disordered structure might be related to fragility.

### 3.5. Temporal Rearrangement of Chorion Components of Alv^−/−^ Eggs

#### 3.5.1. Time-Dependent Changes of Major Chorion Components

It has been shown that self-assembled rearrangements of chorion components occur during the chorion hardening at fertilization and the molecular size changes of major components are well described [15,16,18,29,30,31,32]. The major components of the unfertilized egg chorion were ZPB and ZPC, which were detected at 75 kDa and 48 kDa, respectively, on SDS-PAGE/CBB (Figure 7A, *Unf* lanes). Immediately after fertilization, the 75 kDa ZPB was cleaved to provide cleaved ZPB (clvZPB) at 60 kDa. Simultaneously, ZPC is crosslinked to clvZPB and ZPB to produce dimers at 120 and 140 kDa, respectively. There is also a high molecular mass smear at >210 kDa, which might be highly crosslinked oligomers (HCO). However, these dimer and smear components could not be observed in the resolution gel, probably because these further crosslinked components would cause their insolubility, which might lead to a tough chorion [17,33,34]. These changes were also observed for *Alv*^+/+^ eggs in water (Figure 7A, left). The 60 kDa component (assignable to clvZPB) appeared as one of the major components at 5 mpf, was reduced at 15 mpf, and disappeared at 30 mpf (Figure 7A, left). The 120 kDa and 140 kDa components (assignable to clvZPB-ZPC and ZPB-ZPC, respectively) and the >210 kDa smear (assignable to HCOs) also appeared at 5 mpf, gradually reduced at 30 mpf, and disappeared at 60 mpf (Figure 7A, left). At 5–30 mpf, the 120 kDa clvZPB-ZPC dimer was dominant over the 140 kDa ZPB-ZPC dimer, indicating that most, but not all, ZPB is processed by Alv.

In the chorion of *Alv*^−/−^ eggs in water (Figure 7A, *midde*), the 75 kDa ZPB and 48 kDa ZPC were detected throughout the post-fertilization period up to 90 mpf examined in this experiment, which is different from the case of *Alv*^+/+^ eggs in water. Notably, no 60 kDa clvZPB or 120 kDa clvZPB-ZPC dimer was detected, indicating that no ZPB cleavage occurred. This indicates that the cleavage of ZPB into clvZPB is catalyzed by Alv at the organism level, as previously reported in vitro [15]. In addition, 140 kDa ZPB-ZPC and >210 kDa HCOs were detected after 30 mpf, concomitantly with a slight decrease in the amount of 75 kDa ZPB and 48 kDa ZPC (Figure 7A, *midde*). Compared with the chorion of *Alv*^+/+^ eggs in water, the kinetics of the dimers and HCO formation were at least six times slower in the absence of Alv in vivo. Thus, these data indicate that ZPB cleavage by Alv accelerates dimer formation and subsequent HCO formation.

In the chorion of *Alv*^−/−^ eggs in BSS (Figure 7A, right), the profiles of temporal changes in chorion components were intermediate between those of *Alv*^+/+^ eggs in water and *Alv*^−/−^ eggs in water. If the disappearance of the HCO smear on the SDS-PAGE/CBB reflects heavy crosslinks, then heavy crosslinks occurred in the chorions of the *Alv*^−/−^ eggs in BSS at 90 mpf. In contrast, they occurred at 30–60 mpf in the chorion of *Alv*^+/+^ eggs in water, while they did not occur even at 90 mpf in the chorion of the *Alv*^−/−^ eggs in water (Figure 7A). These data are consistent with the results showing that the chorion toughness for *Alv*^−/−^ eggs in BSS was intermediate between those of *Alv*^+/+^ eggs in water and *Alv*^−/−^ eggs in water (Figure 4C). Interestingly, the conversion of 75 kDa ZPB to 60 kDa clvZPB and the formation of 120 kDa clvZPB-ZPC dimers also occurred in the chorion of *Alv*^−/−^ eggs in BSS similar to the *Alv*^+/+^ eggs in water, irrespective of the absence of Alv at the organism level. These changes were very slow in the chorions of *Alv*^−/−^ eggs in BSS. They were observed at 30 mpf in this case, while at 5 mpf in the chorion of *Alv*^+/+^ eggs in water. These observations suggest that there is a ZPB-processing protease other than Alv; however, this enzyme has not yet been identified. It remains unknown which ions are necessary for the activity in BSS. Nevertheless, its activity did not appear to be effective in cleaving ZPB, since the 120 kDa dimer was less than the 140 kDa dimer at 60 mpf. Analyzed together, these results suggest that Alv is a critical protease for the conversion of ZPB to clvZPB and subsequent crosslinking with ZPC and other chorion components.

#### 3.5.2. Alv-Dependent Processing of the Hardening TGase

It has been shown that TGase activity is important for chorion hardening, i.e., hardening TGase is involved in the crosslinking of clvZPB with ZPC and subsequent HCO formation [34]. Therefore, changes in TGase during chorion hardening were monitored by Western blotting using an anti-hardening TGase antibody (Figure 7B). In the chorions of both *Alv*^+/+^ and *Alv*^−/−^ unfertilized eggs, 65, 56, and 53 kDa components were detected (Figure 7B, *Unf* lanes). It should be noted that the size of a major component, 65 kDa, was slightly smaller than the calculated molecular mass of the cloned medaka hardening TGase, 76 kDa (Yasumasu S., to be published elsewhere). The data suggest that medaka hardening TGase might be proteolytically processed during expression into the chorion. Of these components, only the 65 kDa component remained at 5 mpf. In the chorion of *Alv*^+/+^ eggs in water (Figure 7B, left), at 5 mpf, a 50 kDa component was newly detected, in addition to the 65 kDa component. At 15 mpf, the 65 kDa component decreased, whereas the 50 kDa component increased. The 65 kDa component further decreased and disappeared at 15–30 mpf, whereas the 50 kDa component gradually decreased at 15–30 mpf and disappeared at 90 mpf. In contrast, the 50 kDa component was never detected in the chorions of *Alv*^−/−^ eggs in water or in BSS at 5–90 mpf (Figure 7B, middle and right), suggesting that the 50 kDa component is dependent on the presence of Alv at the organism level. These results indicate that 65 kDa TGase is processed to 50 kDa TGase by Alv. To the best of our knowledge, this is the first study to show that Alv is involved in TGase processing. In addition, the 65 kDa and the processed 50 kDa TGase gradually disappeared at 30 mpf in the chorion of *Alv*^+/+^ eggs in water, whereas in the chorions of *Alv*^−/−^ eggs in water or BSS, the 65 kDa TGase remained at least until 90 mpf without being processed to 50 kDa TGase. These data suggest that, once processed from 65 kDa TGase by Alv, 50 kDa TGase is self-crosslinked and/or crosslinked to other chorion components, which could not be detected in the resolution gel using Western blotting.

## 4. Discussion

Chorion components are already installed in unfertilized eggs during oogenesis and, during egg activation and subsequent cortical reaction, the elevation and hardening of the chorion occurs in a self-assembled manner. Chorion hardening is an important event for fertilization and subsequent development before hatching, complete polyspermy block, and biological and mechanical protection of developing organisms from various risks [35,36]. The self-assembly of the chorion is triggered by the cleavage of a major chorion component ZPB, followed by crosslinking with ZPC and other chorion components to complete chorion hardening [15,16,18,29,30,31,32]. It has been shown that ZPB processing is catalyzed by a proteinase that is secreted from the egg cortical alveoli to the chorion upon fertilization [15,16], while crosslinking is catalyzed by a TGase that pre-exists as a chorion component, with its activity suppressed in unfertilized eggs [33]. In 2000, Alv was purified from cortical alveolus exudates and identified to reproduce chorion hardening in vitro [15], although the TGase remained unidentified. By establishing an *Alv*^−/−^ medaka strain, this study demonstrated for the first time that Alv facilitates chorion hardening in two ways. One is the processing of 75 kDa ZPB into 60 kDa clvZPB (Figure 7A) and the other is the processing of 65 kDa TGase into 50 kDa TGase (Figure 7B). The 50 kDa TGase might be an activated form. The presence of a proteinase for processing the chorionic TGase has been suggested in medaka [32,33] and rainbow trout [25,37]. Notably, both 60 kDa clvZPB and 50 kDa TGase already appeared to be involved in the formation of HCOs at 5 mpf (Figure 7A,B). Thus, Alv is concluded to be the earliest trigger of chorion hardening at the organism level.

Most recently, ZPC-deficient medaka has been reported [38]. ZPC-deficient females spawn string-like materials containing crushed eggs from the genital pore and ZPC-deficient eggs cannot develop even after in vitro fertilization. The egg component ZPC is more important than Alv for the survival of the organism. The *Alv*^−/−^ fertilized eggs have a soft chorion and, although rare, show some polyspermy phenotypes, such as multiple blastodiscs and oil droplet malposition (data not shown). The soft chorion had a female-specific phenotype (Figure 2D). Once hatched successfully, they survive, develop, become reproductive, and produce offspring. Alv exclusively affects chorionic properties. The *Alv*^−/−^ fertilized chorion is a thin and enlarged envelope, which may be related to the low toughness of *Alv*^−/−^ chorion. Notably, although the *Alv*^−/−^ chorion retains a basic multiple-layer structure (Figure 6A,B), multiple porous structures were observed in the periphery of the inner layer of *Alv*^−/−^ chorion (Figure 6C). They must be formed by a stretching force that arises when the chorion is enlarged by osmotic pressure during PVS formation via water absorption. The multiple porous structures of the *Alv*^−/−^ chorion might be related to the permeability to 10 kDa FITC-dextran, which is in contrast to the impermeability of the WT chorion, which has no porous structure (Figure 5A and Figure 6C).

The *Alv*^−/−^ chorion from the egg that is fertilized and cultured in BSS shows higher toughness than that from the egg prepared in water (Figure 4C). The chorion of *Alv*^−/−^ egg in BSS is thicker than that in water (Figure 6C). It is also less enlarged than that in water, which can be deduced from a smaller PVS volume of the egg in BSS than that in water (Figure 3B). The thicker and less enlarged properties of chorion are correlated with higher toughness of *Alv*^−/−^ chorion in BSS than in water. On the other hand, the permeability of the *Alv*^−/−^ chorion to 10 kDa FITC-dextran is the same between the egg in BSS and that in water (Figure 5B). As discussed above, the permeability appears to be related to the disordered, multiple porous structures in the periphery of the inner layer of *Alv*^−/−^ chorion (Figure 6C). Consistent with this, the porous structures are also prominent in *Alv*^−/−^ chorion of the eggs in BSS (Figure 6C, right). We do not know why the chorion properties are different between the egg in BSS and that in water. However, we are hypothesizing that, in addition to Alv, there is another ZPB-processing protease, though not yet identified, that might act on ZPB at 30–60 mpf in BSS to trigger heavy crosslinks of the chorion components (Figure 7A, right). This change might cause a slow facilitation of the toughness of the *Alv*^−/−^ egg at 60 mpf and later in BSS, but not in water (Figure 4B, *lower*). In contrast, This ZPB-processing and the slow facilitation of toughness of the *Alv*^−/−^ egg do not occur in water (Figure 7A, middle, Figure 4B, middle). At any rate, it is interesting to identify an unknown ZPB-processing protease in the future.

This study clarifies the in vivo significance of Alv in chorion hardening. On the other hand, we also observed an interesting phenomenon that may or may not be related to chorion hardening. Thus, the *Alv*^−/−^ fry had significantly lower body length and higher yolk volume than those of the WT at the hatching stage (Figure 1D). These features of hatched fry suggest that the yolk adsorption of fry for their nutrition acquisition is directly affected by the absence of Alv. For example, Alv may be involved in the processing of degrading enzymes of yolk materials, which facilitates yolk absorption during pre-hatching stages. Alternatively, it suggests that embryo and fry development is delayed through indirect effects from the Alv deficiency. Alv affects both the toughness and permeability of the chorion (Figure 4 and Figure 5). The toughness itself may not be essential, because fry can hatch when they are taken care of well. Therefore, the higher permeability of the *Alv*^−/−^ chorion might lose some important factors that the support growth of embryos and fry in the PVS. To demonstrate Alv’s new functions, further studies are necessary in the future.

Since the high permeability of the soft chorion has no effect on the survival of pre-hatching fry at 0–9 dpf, soft chorion eggs can be used for the functional study of PVS components during development [7,39,40] and for a toxicity test of large molecules (ca.10 kDa) in developing embryos [26,41].

## Figures and Tables

**Figure 1 biomolecules-13-00146-f001:**
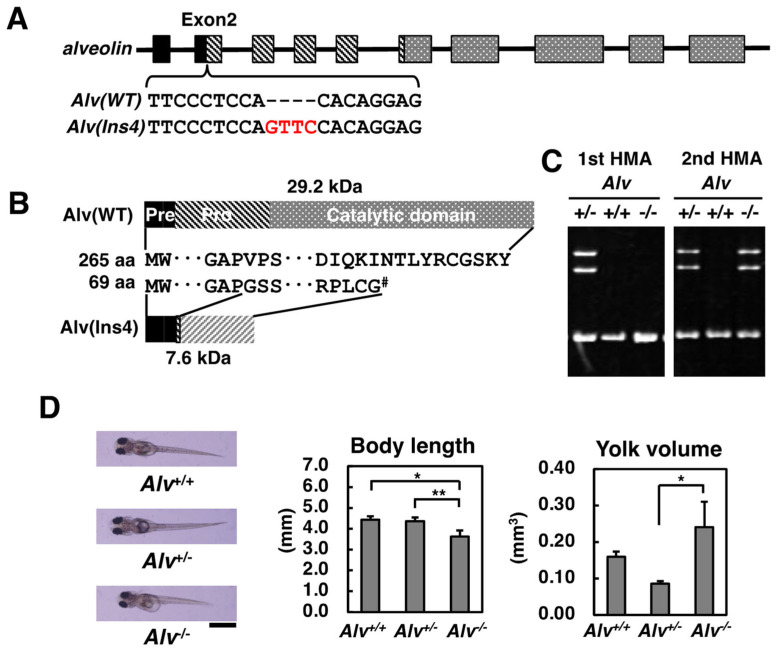
Generation and the pre-hatching development of the *Alveolin*-deficient medaka. (**A**) Exon/intron organization of the *alveolin* gene. *Alv(Ins4)*, a knockout allele due to a 4 bp-insertion (GTTC) in exon 2 colored in red. *Alv(WT)*, the corresponding wild-type allele. The boxes and bars stand for the exon and intron, respectively; (**B**) A 29.2 kDa-polypeptide (265 aa) is synthesized from *Alv(WT)* allele and consists of pre-, pro-, and catalytic domains. A 7.6 kDa-polypeptide (69 aa) is synthesized from *Alv(Ins4)* allele, due to the premature termination codon (#) arisen from the 4 bp-insertion, and is devoid of the catalytic domain; (**C**) Genotyping through two rounds of HMAs. *Alv*^+/+^ and *Alv*^−/−^ are not differentiated at the first HMA but differentiated at the second HMA after mixing with the *Alv*^+/+^ amplicon; (**D**) Phenotype of 9 dpf fry of *Alv*^+/+^, *Alv*^+/−^, and *Alv*^−/−^ genotypes cultured in water. Photos of 9 dpf fry (left) taken from dorsal view. The scale bar, 10 mm. Body length (middle) and yolk volume (right) of fry. Volumes of yolk were calculated as described in Materials and Methods. The number of fry tested was 22, 16, and 4 for *Alv*^+/+^, *Alv*^+/−^, and *Alv*^−/−^, respectively. ** *p* < 0.01, * *p* < 0.05.

**Figure 2 biomolecules-13-00146-f002:**
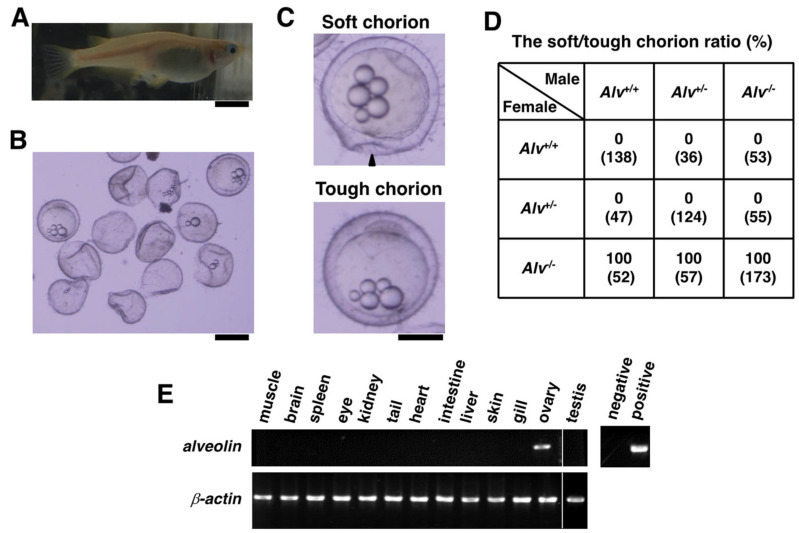
Soft chorion phenotype of the *Alv*^−/−^ eggs. (**A**) Spawned eggs attached at the genital pore of the *Alv*^−/−^ female at 1–2 h-post-fertilization (hpf). The scale bar, 10 mm; (**B**) The spawned eggs attached on the female in (**A**). Most eggs (90%) were crushed and very few could survive. The scale bar, 1 mm; (**C**) *Upper*, A survived *Alv*^−/−^ egg. Its chorion was very soft and deformed when gently pricked with the tip of forceps (arrowhead). *Lower*, a WT egg. The phenotype of *Alv*^−/−^ egg is named “soft chorion”, while that of WT is “tough chorion”. The scale bar, 0.5 mm; (**D**) The soft chorion/tough chorion ratio in reciprocal crossing experiments. The numbers in the parentheses stand for the number of fertilized eggs observed. (**E**) Organ-specific expression of the *alveolin* gene as detected by RT-PCR. The *actin* gene expression was also shown. The size of amplicons is 804 and 840 bp for *alveolin* and *β-actin* genes, respectively. Negative and positive controls were water and a pGEM-alveolin plasmid used as a template, respectively.

**Figure 3 biomolecules-13-00146-f003:**
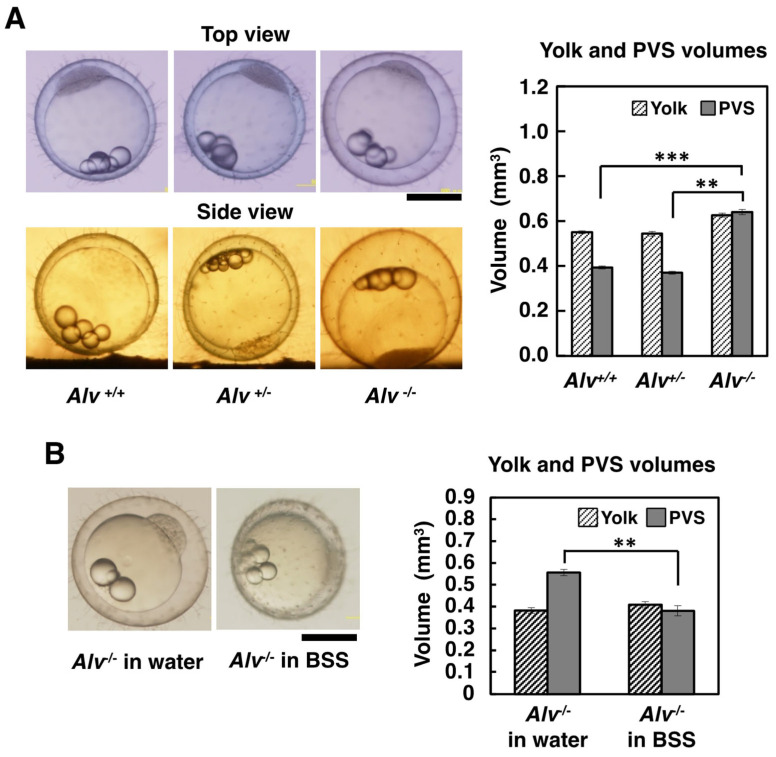
Large egg phenotype of *Alv*^−/−^ embryos at 4 hpf. (**A**) Photos of 4 hpf embryos of *Alv*^+/+^, *Alv*^+/−^, and *Alv*^−/−^ genotypes in water from the top (left upper) and side (left lower) views. The scale bar, 0.5 mm. *Right*, the yolk and PVS volumes, calculated as described in Materials and Methods. The number of embryos tested was 23, 17, and 20 for *Alv*^+/+^, *Alv*^+/−^, and *Alv*^−/−^ genotypes, respectively. ** *p* < 0.01, *** *p* < 0.001; (**B**) The left photos of *Alv*^−/−^ embryos at 4 hpf in water and BSS. The embryo was fertilized and developed in water (Left) or BSS (Right). The scale bar, 0.5 mm. The right bar graph shows the yolk and PVS volumes. The number of embryos tested was 22. ** *p* < 0.01.

**Figure 4 biomolecules-13-00146-f004:**
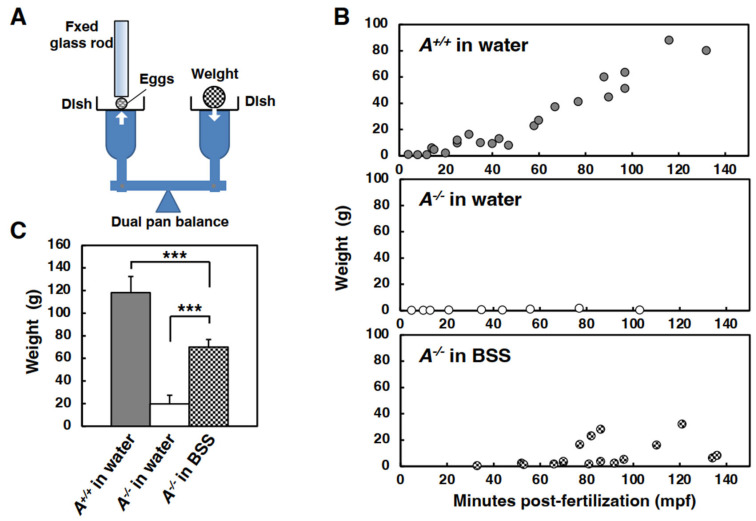
The toughness of chorion of the *Alv*^−/−^ fertilized egg. (**A**) A sketch of the device used for measuring the toughness of chorion. An egg is placed on the left dish and a glass rod is fixed to touch on the egg. When the weight (coin) is put on the right dish, the egg is pressed between the glass rod and the dish. The total weights put on the right dish when the egg is crushed are measured as an indicator. (**B**) Time-dependent changes in the toughness of chorion in the eggs at 0–140 mpf. The weight vs. mpf is plotted for each egg. *Upper*, *Alv*^+/+^ eggs in water; *Middle*, *Alv*^−/−^
*eggs in water*; *Lower*, *Alv*^−/−^ eggs in BSS. (**C**) The toughness of the eggs at 3 hpf. *Alv*^+/+^ eggs in water, *Alv*^+/+^ eggs fertilized and incubated for 3 h in water (*n* = 3); *Alv*^−/−^
*eggs in water*, *Alv*^−/−^ eggs fertilized and incubated for 3 h in water (*n* = 3); *Alv*^−/−^ eggs in BSS, *Alv*^−/−^ eggs fertilized and incubated for 3 h in BSS (*n* = 3). *** *p* < 0.001.

**Figure 5 biomolecules-13-00146-f005:**
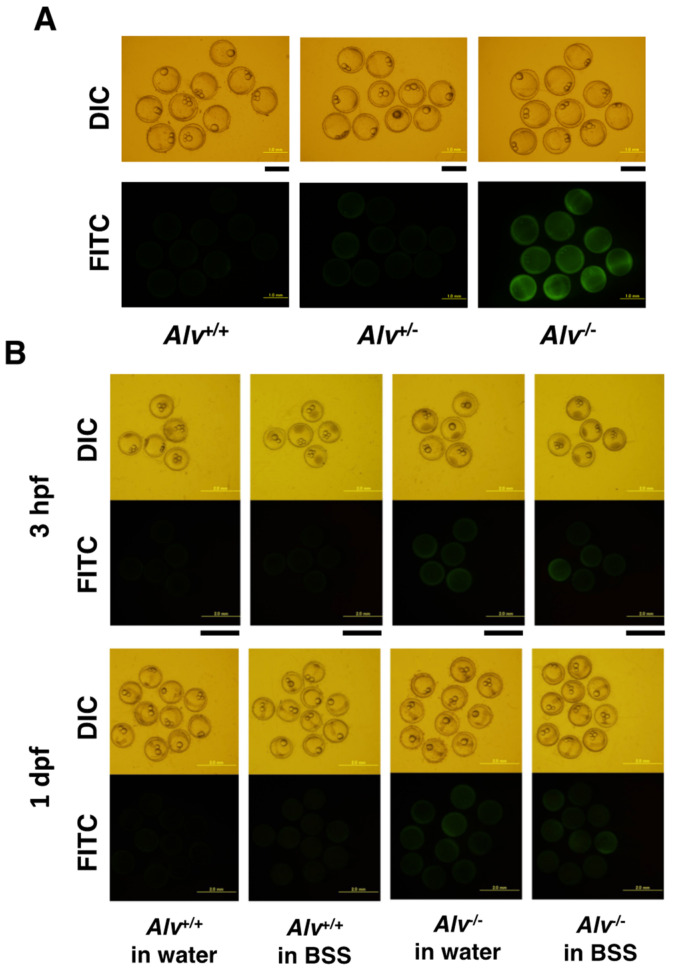
Permeability of the chorions. Images of the fertilized eggs after the incubation with FITC-dextran are shown. The *upper* and *lower* panels are differential interference contrast and fluorescent images, respectively. (**A**) Images of the fertilized eggs at 3 hpf with *Alv*^+/+^ and *Alv*^−/−^ genotypes. *Alv*^+/+^, *Alv*^+/+^ fertilized eggs in water; *Alv*^−/−^, *Alv*^−/−^ fertilized eggs in water. The scale bars, 1 mm; (**B**) Images of the fertilized eggs at 3 hpf and 1 dpf of *Alv*^+/+^ eggs in water, *Alv*^+/+^ eggs in BSS, *Alv*^−/−^ eggs in water, and *Alv*^−/−^ eggs in BSS. The scale bars, 2 mm.

**Figure 6 biomolecules-13-00146-f006:**
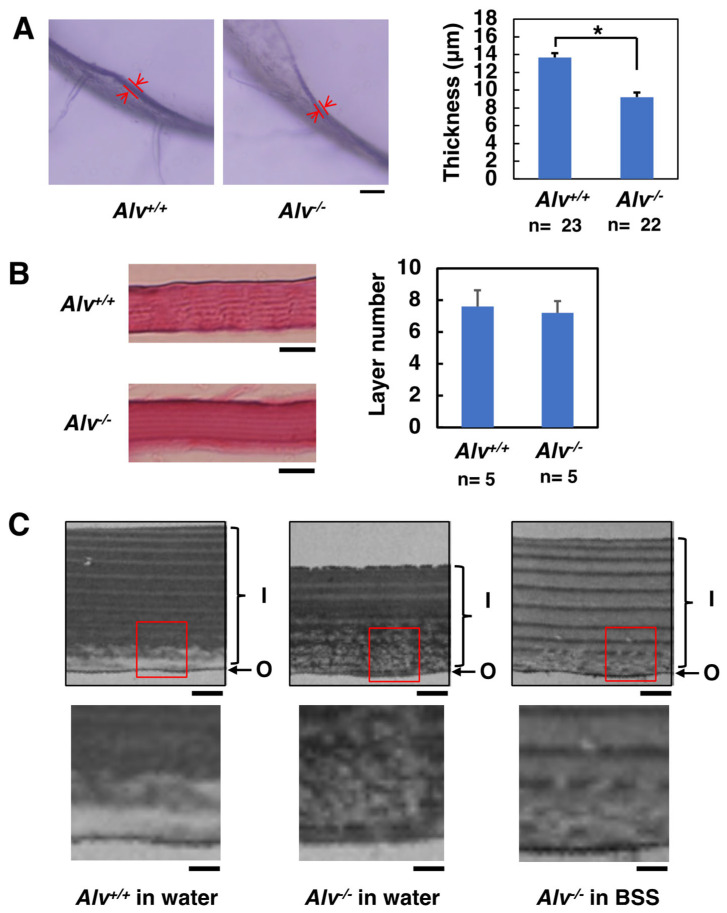
Morphological observation of the chorions. (**A**) Light microscopic images (left) and the thickness (right). Light microscopic images of the chorion. The scale bar, 50 mm. The chorion section indicated by the red arrows was measured as the thickness. The average thickness is shown and the error bars represent standard deviations. * *p* < 0.05; (**B**) Hematoxylin-eosin staining (left) and the layer number (right). Photos of the chorion from the *Alv*^+/+^ and *Alv*^−/−^ fertilized eggs. The scale bars, 10 mm. The average number of the layers at five different parts of fertilization envelope are shown. The error bars represent standard deviations; (**C**) Transmission electron microscopic images. The upper panels show the photos of chorions derived from *Alv*^+/+^ eggs in water, *Alv*^+/−^ eggs in water, and *Alv*^−/−^ eggs in BSS. I, the inner layer; O, the outer layer. The scale bar, 10 mm. The lower panels show magnified images of the red squares in the upper panels. The scale bar, 100 nm.

**Figure 7 biomolecules-13-00146-f007:**
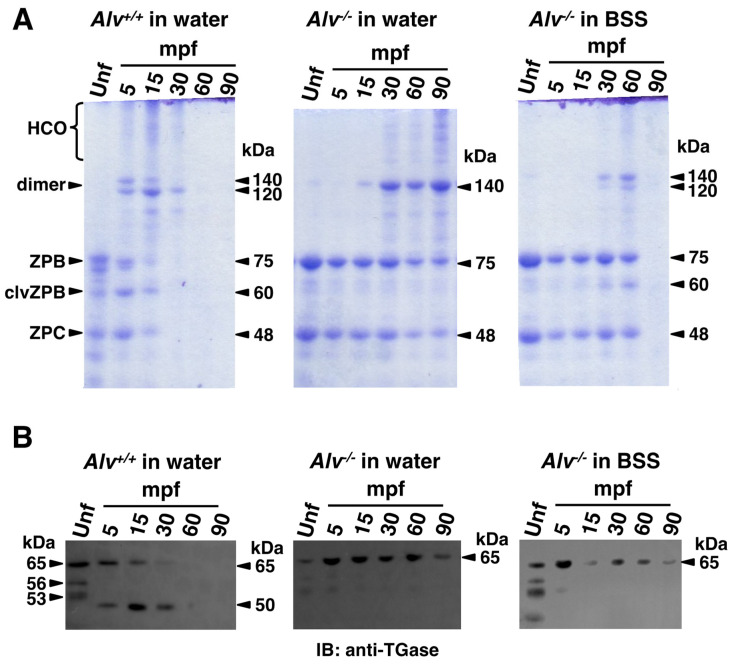
Time-dependent changes of chorion components after fertilization. (**A**) SDS-PAGE/CBB staining of the chorions isolated from the unfertilized eggs (Unf) and the fertilized eggs at 5, 15, 30, 60, and 90 min post-fertilization. *Alv*^+/+^ in water, *Alv*^+/+^ eggs fertilized and incubated in water; *Alv*^−/−^ in water and *Alv*^−/−^ in BSS, *Alv*^−/−^ eggs fertilized and incubated in water and in BSS, respectively. ZPB (75 kDa), cleaved ZPB (60 kDa), and ZPC (48 kDa) were assigned as described in the text. (**B**) Western blotting of the chorion preparations as described in (**A**), using anti-hardening transglutaminase (TGase) antibody. Hardening TGase (65 kDa) and the processed TGase (50 kDa) are assigned as described in the text.

## Data Availability

Data is contained within this article.

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
