# Peer review of "Critical Role of the Cortical Alveolus Protease Alveolin in Chorion Hardening In Vivo at Medaka Fertilization"

_biomolecules, 2023, doi:10.3390/biom13010146_

Round 1

Reviewer 1 Report

I read with pleasure and interest this article “Critical role of the cortical alveolus protease alveolin in chorion hardening in vivo at medaka fertilization”. I found the work well written and very detailed.
I must admit that the numbers of individuals (fish), eggs, embryos and fry used in the experiments had puzzled me when reading the materials and methods. Still in the materials and methods, it should be included from reproductive performance data (ie. fertilization and hatching rates). Moreover, the statistics methods should be better written, including normality tests, whether it were used a software or not. About the discussion, I found the discussion well written, but it is too short and missing coments about some results, like morphological observations of the chorion of Alv-/- eggs, permeability.
Consequently, I believe the work can be published in the journal after some adjust in the manuscript.

Author Response

To: Reviewer 1 

I read with pleasure and interest this article “Critical role of the cortical alveolus protease alveolin in chorion hardening in vivo at medaka fertilization”. I found the work well written and very detailed.

(Reply) We are pleased to hear that you find merits in our manuscript.  

I must admit that the numbers of individuals (fish), eggs, embryos and fry used in the experiments had puzzled me when reading the materials and methods. Still in the materials and methods, it should be included from reproductive performance data (ie. fertilization and hatching rates).

(Reply) Thank you for valuable comments on the numbers of fish, eggs, embryos and fry in “materials and methods” (M&M) part. We tried to collect as many medaka samples as possible for each genotype to evaluate the statistical significance, and actually described their numbers in the figure legends. However, we did not describe reproductive performance data in M&M. Therefore, we added necessary information in M&M, which might be useful for better understanding of our experiments:

For the section “2.6” (New lines 141-144), we have added the following sentences: “The Alv+/- medaka were mated to obtain Alv+/+, Alv+/-, and Alv-/- embryos at 4 hpf. The fertilization rate was 90-100%, and the hatching rate was 90-100% for Alv+/+ and Alv+/- fry, and around 20% for Alv-/- fry on the culture in water. The genotyping was performed after morphological observations”. 

We have also specified the developmental stages of medaka in the sections 2.6 to 2.8 in M&M for clarity: “4 hpf embryos and 9 dpf fry just after hatching” for section 2.6 (New lines 144-145); “The egg at 0-140 minutes post-fertilization (mpf)” for section 2.7 (New line 157); “Eggs at 3 hpf and 1 dpf with Alv+/+ and Alv-/-genotypes” for section 2.8 (New line 163).  

Moreover, the statistics methods should be better written, including normality tests, whether it were used a software or not.

(Reply) Thank you for your important point-out. We have added some more information about “statistical analysis” in the M&M” as follows (New lines 224-226):

“Data were processed to show the mean and the standard deviation for normality of the data set. Statistical analysis was performed using Student’s t-test. Statistical significance was set at p < 0.05. All the data were processed using the Excel software”.

About the discussion, I found the discussion well written, but it is too short and missing coments about some results, like morphological observations of the chorion of Alv-/- eggs, permeability.

(Reply) Thank you for your comments on the discussion. We already discussed the TEM images in relation to the permeability of the Alv-/- eggs (New Lines 549-555). However, no discussion has been done about differences in chorion properties between fertilized eggs in BSS and those in water. According to your comments, we have added the following discussion as the third paragraph (New lines 556-574):

“The Alv-/- chorion from the egg which is fertilized and cultured in BSS shows higher toughness than that from the egg prepared in water (Figure 4C). The chorion of Alv-/- egg in BSS is thicker than that in water (Figure 6C). It is also less enlarged than that in water, which can be deduced from a smaller PVS volume of the egg in BSS than that in water (Figure 3B). The thicker and less enlarged properties of chorion are correlated with higher toughness of Alv-/- chorion in BSS than in water. On the other hand, the permeability of the Alv-/- chorion to 10 kDa FITC-dextran is the same between the egg in BSS and that in water (Figure 5B). As discussed above, the permeability appears to be related to the disordered, multiple porous structures in the periphery of the inner layer of Alv-/- chorion (Figure 6C). Consistent with this, the porous structures are also prominent in Alv-/- chorion of the eggs in BSS (Figure 6C, right). We do not know why the chorion properties are different between the egg in BSS and that in water. However, we are hypothesizing that, in addition to Alv, there is another ZPB-processing protease, though not yet identified, that might act on ZPB at 30-60 mpf in BSS to trigger heavy crosslinks of the chorion components (Figure 7A, right). This change might cause slow facilitation of toughness of the Alv-/- egg at 60 mpf and later in BSS, but not in water (Figure 4B, lower). In contrast, This ZPB-processing and the slow facilitation of toughness of the Alv-/- egg do not occur in water (Figure 7A, middle, Figure 4B, middle). At any rate, it is interesting to identify an unknown ZPB-processing protease in the future.”

Reviewer 2 Report

The authors generated a gene knockout line for medaka to investigate the functions of Alv in mediating chorion hardening. They conducted well designed morphological, histological and biochemical experiments to characterize effects of Alv deficiency on PVS enlargement, toughness and permeability of egg chorion. They proved that Alv plays important roles in chorion hardening and identified for the first time that Alv is also involved in processing the TGase during chorion hardening. The results shed new light on the functions and mechanisms of Alv in fish embryogenesis and reproduction. I have the following concerns.

Major

Biological significance of fish chorion hardening should be discussed to further understanding of the molecular functions of the medaka Alv protein. The authors indicate that the alv-/- larvae had significantly lower body length and higher yolk volume than those of the wild type (Figure 1D). However, yolk volume of the 4-hpf embryos of the mutant was not significantly different from that of the WT (Figure 3A). These results suggest that Alv of medaka also has functions in embryo and larva development except for mediating chorion hardening. Are these developmental defects the direct results of impaired chorion hardening? Although it is not the theme of this study, the possible mechanism should be discussed.

PVS enlargement, decrease of toughness and increase of permeability of egg chorion of the Alv-/- mutants could be partially rescued by fertilizing and incubating the eggs in BSS medium. It is likely that some ingredients of BSS have a function in regulating chorion hardening independent of Alv. However, the authors didn’t mention both composition of BSS and the counterpart enzymes that have a Alv-like function.

Minor

Figure 1D, direction of the specimens should be indicated. Dorsal view of lateral view?

Line 35, that is named differently named

Lines 70-71, I couldn’t find any data supporting the statement that Alv has effects on egg spawning behavior.

Line 137, perivitteline

Line 249, When spawned and fertilized eggs were the focus, severe morphological damage was observed. Hard to understand, consider revision.

Figure 2E, the gene names should be unified in the figure and the corresponding legend.

Lines 303 and 307, The scale bar, 500 mm.

Author Response

To: Reviewer 2

Biological significance of fish chorion hardening should be discussed to further understanding of the molecular functions of the medaka Alv protein. The authors indicate that the alv-/- larvae had significantly lower body length and higher yolk volume than those of the wild type (Figure 1D). However, yolk volume of the 4-hpf embryos of the mutant was not significantly different from that of the WT (Figure 3A). These results suggest that Alv of medaka also has functions in embryo and larva development except for mediating chorion hardening. Are these developmental defects the direct results of impaired chorion hardening? Although it is not the theme of this study, the possible mechanism should be discussed. 

(Reply) Thank you for pointing out interesting phenotypes of Alv-/- larva that were not discussed in the original manuscript. Yes, we are also wondering why the Alv-/- had significantly lower body length and higher yolk volume than those of the WT at the hatching stage (Figure 1D). We agree to your interpretation that Alv might also have functions in embryo and larva development in addition to chorion hardening. We have some thoughts on this, but we avoided writing them because they were too speculative at this stage. However, according to your suggestion, we have added the following sentences as the fourth paragraph (New lines 575-590):

“This study clarifies the in vivo significance of Alv in chorion hardening. On the other habd, we also observed an interesting phenomenon that may or may not be related to chorion hardening. Thus, the Alv-/- fry had significantly lower body length and higher yolk volume than those of the WT at the hatching stage (Figure 1D). These features of hatched fry suggest that yolk adsorption of fry for their nutrition acquisition is directly affected by the absence of Alv. For example, Alv may be involved in processing of degrading enzymes of yolk materials, which facilitates yolk absorption during pre-hatching stages. Alternatively, it suggests that embryo and fry development is delayed through indirect effects from the Alv deficiency. Alv affects both toughness and permeability of the chorion (Figure 4, 5). The toughness itself may not be essential, because fry can hatch when they are taken well care of. Therefore, higher permeability of the Alv-/- chorion might lose some important factors that support growth of embryos and fry in the PVS. To demonstrate Alv’s new functions, further studies are necessary in the future.”  

PVS enlargement, decrease of toughness and increase of permeabity of egg chorion of the Alv-/- mutants could be partially rescued by fertilizing and incubating the eggs in BSS medium. It is likely that some ingredients of BSS have a function in regulating chorion hardening independent of Alv. However, the authors didn’t mention both composition of BSS and the counterpart enzymes that have a Alv-like function. 

(Reply) Thank you for valuable comments. For BSS, we have added the composition in the section “2.1” (New lines 80-81): the salt solution contains 0.1 M NaCl, 5 mM KCl, 1 mM CaCl2, and 0.6 mM MgSO4.

For the counterpart enzymes that have a Alv-like function, we already described in the results (New lines 483-484) as “there is a ZPB-processing protease other than Alv; however, this enzyme has not yet been identified.”  We have added the following sentence right after the sentence (New lines 484-485): “It remains unknown which ions are necessary for the activity in BSS.”

Figure 1D, direction of the specimens should be indicated. Dorsal view of lateral view?

(Reply) We have added the phrase “taken from dorsal view.” (New line 248).

Line 35, that is named differently named

(Reply) that is differently named  (New line 35)

Lines 70-71, I couldn’t find any data supporting the statement that Alv has effects on egg spawning behavior.

(Reply) Thank you for the comment. The phrase “and the subsequent egg spawning behavior” has been removed (New line 70-71).

Line 137, perivitteline

(Reply) “perivitelline” (New line 140)

Line 249, When spawned and fertilized eggs were the focus, severe morphological damage was observed. Hard to understand, consider revision. 

(Reply) Thank you for the comments. The sentence in question has been deleted, because it is not necessary. (New line 265)

Figure 2E, the gene names should be unified in the figure and the corresponding legend.

(Reply) In Figure 2E: Actin and Alveolin have been changed to b-actin and alveolin, respectively. The revised Figure 1 has been inserted instead of the old version.

Lines 303 and 307, The scale bar, 500 mm.

(Reply) Thank you for the comment. In Figure 3B legend, “500” mm should read “0.5” mm. (New lines 322 and 326)